# Internet Use and Perceived Parental Involvement among Adolescents from Lower Socioeconomic Groups in Europe: An Exploration

**DOI:** 10.3390/children10111780

**Published:** 2023-11-02

**Authors:** Roy A. Willems, Peter K. Smith, Catherine Culbert, Noel Purdy, Jayne Hamilton, Trijntje Völlink, Herbert Scheithauer, Nora Fiedler, Antonella Brighi, Damiano Menin, Consuelo Mameli, Annalisa Guarini

**Affiliations:** 1Department of Psychology, Open Universiteit, 6419 AT Heerlen, The Netherlands; trijntje.vollink@ou.nl; 2Department of Psychology, Goldsmiths, University of London, London SE14 6NW, UK; p.smith@gold.ac.uk (P.K.S.); c.culbert@lancaster.ac.uk (C.C.); 3Centre for Research in Educational Underachievement, Stranmillis University College, Belfast BT9 5DY, UK; n.purdy@stran.ac.uk; 4School of Social Sciences, Education and Social Work, Queen’s University, Belfast BT7 1PS, UK; jpatterson43@qub.ac.uk; 5Department of Education and Psychology, Freie Universität Berlin, 14195 Berlin, Germany; hscheit@zedat.fu-berlin.de (H.S.); nora.fiedler@fu-berlin.de (N.F.); 6Faculty of Education, Free University of Bolzano, 39042 Bolzano, Italy; antonella.brighi@unibz.it; 7Department of Humanities, University of Ferrara, 44121 Ferrara, Italy; damiano.menin@unife.it; 8Department of Education Studies “Giovanni Maria Bertin”, University of Bologna, 40126 Bologna, Italy; consuelo.mameli@unibo.it; 9Department of Psychology “Renzo Canestrari”, University of Bologna, 40127 Bologna, Italy; annalisa.guarini@unibo.it

**Keywords:** Internet use, adolescents, socioeconomic disadvantage, parents, online, apps

## Abstract

Internet usage is a salient developmental factor in adolescents’ lives. Although relevant correlates of Internet use have been documented earlier, there is a lack of information on lower socioeconomic status groups. This is important, as these adolescents have increased risk of negative online experiences. The current survey aimed to explore Internet use and parental involvement amongst adolescents from areas of socio-economic disadvantage in 30 urban schools across five European countries. A total of 2594 students participated, of whom 90% were 14–16 years. Virtually all adolescents of socioeconomic disadvantage had Internet access, with 88.5% reporting spending more than two hours per day online, often on apps such as Instagram, Snapchat, and YouTube. Almost one-third of adolescents did not talk with their parents about their Internet use and almost two-thirds indicated that their parents were only a little or not interested in their Internet use. A consistent finding across countries was that girls more often talked with their parents about their Internet use and more often reported that their parents were interested in their Internet use than boys. The results suggest that parents have an important task in explicitly showing interest in their adolescents’ Internet use, with special attention needed for boys.

## 1. Introduction

Internet use has increased at a very rapid rate this century, and noticeably in young people [1,2] and on social networking sites. Although the average age restriction used by social networking sites is 13 years of age, younger children are increasingly engaging with these sites [3,4]. A survey in England by Hayes et al. [4], with a sample of 883 children aged 7 to 12 years, found that 40% identified accessing social networking sites, with about half of whom doing so once a week and half about once a day. However, usage increases greatly in adolescence; in the UK, Ofcom [3] reported in 2018 that 99% of 12- to 15-year-olds went online, for an average of 20.5 h per week; 83% had their own smartphone, and 50% their own tablet. A survey in Europe carried out in 2018–2019 [5] found that the majority of children aged 9 to 16 years in 19 European countries report using their smartphones ‘daily’ or ‘almost all the time’ and the time children now spend online varied between about two to three-and-a-half hours per day.

Internet usage is recognized as an important developmental factor in the lives of adolescents. In a narrative review on adolescents, screens, and social media, Orben [2] summarized that there are links between Internet and social media use to wellbeing, on average negative but small, with many potential confounders. In addition, the association between extent of Internet use and wellbeing may be curvilinear (the ‘Goldilocks effect’) rather than linear [6]: a moderate amount of use may help adolescents take up the opportunities provided by the Internet and develop skills and for coping with the risks involved. Nevertheless, excessive Internet use is widely seen as a risk factor for cyberbullying and other harmful activities. Problematic social media use has been reported in 4.5–9.4% of European adolescents [7].

Socially disadvantaged youth is a vulnerable group with respect to risky or addictive behavior, and it is therefore important take into account specific factors linked to youth in structurally disadvantaged regions when developing health programs [8]. With respect to Internet use, several studies point out that problematic Internet use is a risk factor in particular for adolescents from areas of socio-economic disadvantage [8,9,10]. For example, the UK Children Go Online project [11] found that children and adolescents from more advantaged homes were more likely to use the Internet and to use it more often and for longer. Additionally, they were more skilled at using the Internet, and had higher levels of self-efficacy regarding use. However, a more recent survey by Harris et al. [12] of Australian children between 6 and 17 years of age found more differences by SES in type rather than amount of use. Participants from higher SES neighborhoods were more exposed to school computers, and participants from lower SES neighborhoods were more exposed to TV, electronic games, mobile phones, and non-academic computer activities at home. Such findings point to the importance of focusing on the kinds of Internet use by adolescents from areas of socio-economic disadvantage and factors associated with that use, which are relatively unexamined [5].

Parental concern and involvement in their child’s Internet use is an important issue with respect to adolescents’ healthy and safe Internet use [13,14]. As parental involvement can be a protective factor in safe Internet use, it can also be a risk factor. Parents can hold active (e.g., supervising or monitoring), evaluative (e.g., talking about online risks) or restrictive strategies (e.g., setting rules or restricting access), and choosing the best combination of strategies can be very challenging [13,15]. For example, when adolescents perceive their parents as being too restrictive, they might search for a way to bypass these restrictions, which might lead to more risky Internet use [16]. Moreover, the role of parents and the application of effective strategies might differ between boys and girls. For example, Wright [17] indicates that a restrictive strategy might be more effective for girls than for boys. In addition, Baldry et al. [13] indicate boys perceive their parents to be less involved and supportive when they are a victim of cyberbullying, while girls reported their parents being more actively involved when being victimized. Another risk factor in parental involvement might be the educational level of the parents. A large European study found an association between lower parental educational level and problematic Internet use [18]. The authors suggest that this relationship might be explained by less educated parents in general being less involved in their child’s Internet use.

### 1.1. Factors Associated with Internet Use and Parental Involvement

Several factors might be associated with adolescents’ (problematic) Internet use and their parents’ involvement, including gender [5], disability [19,20], ethnicity [21], and school performance [22]. First, Wright et al. [23] suggest that boys are more at risk of developing problematic Internet use than girls. However, there is still inconsistency in the literature. For example, the EU Kids Online follow-up [5] reported that ‘for the most part, gender differences are few and far between. Where they do exist, they are often inconsistent in ways that are difficult to explain, with girls having slightly more of some kinds of online experiences than boys in one country, but with the reverse finding in another country’ (pp.132–133). Concerning parental involvement—as indicated before—it is suggested that parental involvement differs with respect to boys and girls, with parents more explicitly being involved with the online experiences of girls, while providing boys with more general information or monitoring in the background [24,25].

Second, the UK Children Go Online project found lower levels of Internet access at home for disabled children [11]. Internet access for children with disabilities provides opportunities to learn, explore, and develop, while on the other hand parents also see risks in Internet usage [20]. This concern is realistic, as children with a disability perceive more cyberhate and are more often a victim of cyberbullying than children without a disability [26,27]. Active parental involvement is therefore of extra importance for this group [27,28].

Third, while the UK Children Go Online project found no specific differences in Internet use in ethnic minority groups [11], these groups have been found as more vulnerable to negative online experiences including racially charged experiences of racial or ethnic discrimination [21,26]. Research on whether ethnic minority groups are also more likely to be victims of cyberbullying is mixed [29], although recent research suggests that they indeed are more likely to be victimized than White youth [30]. Again, active parental involvement is of importance for this group to address negative online experiences they encounter [30].

Finally, school performance is an important factor related to Internet use. Research has found a relation between problematic Internet use and academic performance, including neglect of schoolwork, absenteeism, and decline in official and self-reported grades [23]. For example, Rozgonjuk and Täht [22] found that using the Internet at school for longer than 60 min per day was associated with decreasing performance in cognitive ability tests and academic outcomes; these were apparent even when controlling for students’ socioeconomic status and gender.

### 1.2. The Blurred Lives Project

The current study was conducted as part of the *Blurred Lives* project (2017–2020; https://www.ou.nl/web/blurred-lives, accessed on 2 November 2023). This Erasmus+ funded project was designed to investigate Internet use amongst adolescents (aged 14–16) from areas of socioeconomic disadvantage across five European countries: Northern Ireland, England, Germany, Italy, and the Netherlands. The project consisted of two phases: in the first phase, students were recruited from a range of secondary schools in areas of socioeconomic disadvantage in each of the five partner countries. They completed an online survey, providing demographic information and aspects of Internet use, reported here. Furthermore, the survey explored any negative experiences they may have had while using the Internet, whether as victims, bystanders, or perpetrators. A second phase of the project used a mixed-methods, co-participatory approach to involve adolescents by training and empowering them to use their voice as co-researchers to help tackle cyberbullying in their peer groups and schools [31,32]. This article reports findings from the first phase on aspects of Internet use and perceived parental involvement in relation to demographic characteristics, across the five participating countries.

While we have some insight in adolescents’ Internet use and relevant factors related to adolescents’ Internet use, there is little insight into how this applies for the at-risk group of adolescents of socioeconomic disadvantage. Furthermore, we have little insight into how this group perceives parental involvement in their Internet use and which demographics are most importantly related to this perception. Therefore, the aim of this explorative study is two-fold. First, we will describe the extent and frequency of Internet use in areas of socioeconomic disadvantage—including popular apps and sites, as well as perceived parental involvement in adolescents’ Internet use. Relevant demographics, including gender, ethnicity, disability, school performance, and economic status will also be included in this description. Second, we aim to explore which demographics correlate most strongly with Internet use and perceived parental involvement. In this large-scale study reporting data from five European countries, we specifically looked for consistent and inconsistent findings across countries. However, we did not use statistical tests to examine country differences, since specific measurement-theoretical conditions should be in place for such comparisons to be made [33,34].

## 2. Materials and Methods

### 2.1. Participants and Procedure

The study was conducted in agreement with ethical guidelines for protecting human participants. All partners obtained approval of their local Ethics Committee. In each of the five countries, schools were selected as being in urban areas and admitting children with above-average levels of socioeconomic disadvantage (i.e., free school meals in Northern Ireland and England; parental income according to the Berlin social structure atlas in Germany; vocational education in The Netherlands and Italy). However, there were slight differences in sampling procedures. For example, Italy included schools with high rates of participants with special educational needs. Otherwise, England offered a book voucher as an incentive for schools to participate. A total of 30 schools agreed to participate and provided informed consent. In Northern Ireland (7 schools) and Italy (3 schools) there was a convenience sampling with no refusals. In England (5 schools), Germany (6 schools), and the Netherlands (9 schools), however, some 50+ schools were contacted initially.

Informed consent from a school representative was required in order to collect data within schools. Furthermore, parents were informed about the research and—depending on the ethical guidelines of the country—had to provide active informed consent. Parents could withdraw permission for their child to participate at any time. Students had to sign informed consent before filling in the questionnaire. In total, 2655 students filled in the questionnaire (age range = 10–26 years). We primarily aimed for grade years containing 14- to 16-year-olds. In total, 2382 students fell within this age range (89.7%). Since we recruited at grade level, some early students (age = 13, *n* = 137) and late students or repeaters (age = 17–18, *n* = 75) also filled in the questionnaire. We decided to include these students in the analyses. A total of 61 pupils were removed from the dataset for not meeting this age criteria of 13–19 years (age < 13, *n* = 7; or age > 18, *n* = 9) or information on age was not entered correctly (i.e., age = 1–4, *n* = 17) or missing (*n* = 28), resulting in a sample size of 2594 students. There were no other exclusion criteria besides the age criterium.

Surveys were conducted between May 2018 and March 2019, usually in the ICT suites of the schools, online; one school in England, however, requested a paper version. The survey was carried out during a normal school lesson time, and typically took some 15–30 min to complete, well within the duration of one lesson period. Students were reminded of the voluntary nature of the survey, and that they could withdraw at any time or not answer certain questions if they wished. One or two teachers and/or research staff members were present, and available to answer questions. Guidance was given to the participants and teaching staff by the research teams and where teaching staff only was present, key information and goals were shared beforehand. Some schools requested immediate feedback on the project, which was always arranged. All schools were invited to a ‘multiplier event’ to disseminate the findings, some months after the survey.

### 2.2. Measurements

The measurements described were part of a larger survey on adolescents’ online experiences. This survey was initially piloted among a small group of adolescents in Northern Ireland, England, and Italy. Translations into Dutch, German, and Italian were checked or back-translated by fluent or native English speakers. Only relevant measures for this article will be described here. These comprised demographic characteristics, aspects of Internet use, and aspects of parental involvement in Internet use.

#### 2.2.1. Demographic Characteristics

We asked about age (years), gender (male/female/other/do not wish to say), residence (where do you live? urban/rural), ethnicity (see below), disability (sensory, physical, learning), how many books or e-books do you have in the house or place you usually live in? (just a few/about 10 or 20/more than 20), and at school, what kind of grades do you usually get? (above average, average, below average). Concerning gender, the category ‘do not wish to say’ was merged with the ‘other’ category, comprising those who might not feel comfortable stating their gender identity in the binary system.

The way ethnicity was measured differed according to the regulations regarding reporting ethnicity/race in each country. Adolescents were asked which country they were born in, which language they spoke at home, and what their ethnic origin was. Based on these questions, ethnicity was then recoded into *majority/minority group membership* status. For Northern Ireland, England, and the Netherlands, pupils were assigned to the majority group when their indicated ethnic background was ‘white’ (or ‘Dutch’ in the Netherlands), spoke the national language at home, and were born in that home country. The Italian team was not allowed to ask for ethnic background and instead asked for country of origin of parents; if one or two of the parents were born in Italy, the teen themselves were born in Italy, and Italian was spoken at home, then the pupil was assigned to the majority group. For Germany, only data on country of birth and language spoken at home were available; students born in Germany were coded as the majority group, unless they were born in another country then moved to Germany and/or spoke another language at home. Cases which deviated from these rules were discussed and resolved within the research team.

The number of books at home served as a proxy of socioeconomic status. Such a measure has been used in PISA surveys [35] (p. 216), and has some empirical validation as an index of economic social and cultural status [36,37].

#### 2.2.2. Aspects of Internet Use

First, pupils were asked about devices owned (do you have your own smart phone/tablet/games console?), whether they had access to other smart phone/tablet/games consoles linked to the Internet at home (yes/no), and whether they had access to the Internet outside home (yes/no). They were then asked two questions on Internet use, being asked to choose one response. *Frequency of use* was asked using the following item: ‘On a usual day, how many times do you go on to the Internet (from school, from home, from an Internet point, hotspot or Wi-Fi, from a mobile or from a tablet or other device)’; with 4 response choices: I don’t, or maybe once/2 to 5 times/about 10 times/more than 10 times. *Time spent on the Internet* was asked with: ‘On a usual day, how much time do you actively spend on the Internet? (e.g., using apps, browsing websites, social media, online gaming, using music/video streaming services); with 5 response choices: I don’t, or less than half an hour/about an hour/about 2 h/3 to 5 h/more than 5 h.

In an open question, pupils were asked about their favorite apps and websites: ‘If you go online, which apps do you use and which sites do you often go to?’ Answers were coded by research teams in England and Northern Ireland into separate categories and these proposed categories were merged into a single coding scheme. This scheme was tested in the other countries and adjusted again to fit each country’s data.

#### 2.2.3. Aspects of Parental Involvement in Internet Use

Pupils were asked three questions about parental involvement in their Internet use. The first concerned communication about Internet usage: ‘Do you tell your parents/carers about which sites you go to online?’ with 4 response choices: always/often/sometimes/no. The second question asked about parental interest in the pupils’ Internet usage: ‘Are your parents/carers interested in your experiences online?’ with 4 response options: yes, they are very interested/they are somewhat interested/they are a bit interested but not very much/no, they are not interested. The third question concerned a multiple-choice question asking about parental attitude about their child’s Internet usage: ‘What do your parents/carers think about your use of the Internet?’, with four response options: they are quite okay about it/they are sometimes concerned about the sites I visit/they are sometimes concerned about the time I spend on the Internet/they are very worried about it and we argue about it.

### 2.3. Statistical Analyses

All analyses were conducted using SPSS version 28. First, descriptives are provided on demographics and Internet use. Findings are presented for the whole sample and separately by country, with the aim of examining similarities and differences. Regarding frequency of Internet use, the first two categories (‘I don’t or maybe once’ and ‘2 to 5 times’) were merged for analysis due to relatively low percentages. Similarly, for time spent, the first two categories (‘I don’t or less than half an hour’ and ‘an hour’) were merged for analysis.

Second, we were interested in which demographic characteristics correlated most strongly with Internet usage and parental involvement. More specifically, we tested whether gender, ethnicity, disability, number of books at home, and grades correlated with the following interval variables: frequency of Internet use, time spent on the Internet, frequency of telling parents about visited sites, parental interest on Internet use, and whether parents were okay about the adolescents’ Internet use. For the latter outcome, we created a binary variable with parents being okay vs. not okay about their Internet use. These analyses were conducted with four linear regression models and one binominal regression model for each country. School was not included as random intercept due to too few clusters per country. Here, gender was coded into dummy variables, with ‘male’ as the reference category. Age, residence, and Internet access were not included in the model due to expected low variation.

## 3. Results

### 3.1. Demographic Data

Table 1 describes the demographic characteristics for each country. Overall, the mean age was 14.77 years (*SD* = 0.88), and this was similar across countries. For gender, 53.9% were male, 42.3% female and 3.9% indicated ‘other’ or did not wish to say. The main gender discrepancy was in Italy, where two-thirds of respondents were male, which can be explained by a majority of males in Italy choosing vocational education. Most pupils came from urban areas, except for the Netherlands; cities in the Netherlands are generally smaller in comparison to the other countries, and many pupils from neighboring towns or villages travel to the city to go to school. Overall, 62.7% of pupils were from the majority group, especially in Northern Ireland and the Netherlands. The major discrepancy was in England: in London, the racial composition, especially of young people in schools, is very diverse, and schools in less advantaged areas often have very high percentages of BAME (Black and Minority Ethnic) pupils. There was a high percentage of children with disabilities in the sample, which is consistent with the aim of selecting schools from disadvantaged areas. On average, 19.3% of pupils indicated having a learning disability, 11.6% a sensory disability, and 4.1% a physical disability.

When pupils were asked about the number of books in their house, on average 31.2% indicated just a few, 20.6% about 10 or 20, and 48.2% more than 20. Here the outlier was Northern Ireland with relatively fewer books being reported. Regarding school grades, 36.1% reported these as good/above average, 57.0% as about average and 7.0% as not so good/below average; few pupils in the Netherlands reported below average grades.

### 3.2. Internet Usage and Parental Involvement

Table 2 gives details about Internet usage. In total, 99.4% of the pupils indicated they had access to an Internet-connected device within the home. In terms of ownership, 97.0% indicated owning a smartphone, 56.2% a tablet, and 69.8% a games console. In total, 78.9% of the pupils had Internet access outside of their home.

When asked how many times pupils go on to the Internet on a usual day (from school, from home, from an Internet point, from a mobile, or from a tablet or other device), 3.4% responded they did not or maybe just once, 18.1% indicated 2 to 5 times, 19.4% about 10 times, and 59.1% more than 10 times a day. When asked about daily time spent online, 2.7% indicated none or less than half an hour, 8.7% about an hour, 18.9% about 2 h, 33.3% 3 to 5 h and 36.3% more than 5 h a day. There was a substantial correlation between frequency of Internet use and daily time spent on Internet (*r* = .48).

In an open question, pupils were asked about the apps they used and sites they visited often. Table 3 shows the list of self-reported apps and websites used per county. Across countries, the top four mentioned apps were Instagram (57.8%), YouTube (49.1%), Snapchat (41.4%), and Facebook (18.0%). While Instagram and YouTube were popular apps in all countries, the popularity of apps differed (see Table 3). For example, while 58% of pupils in Northern Ireland mentioned Facebook as a frequently used app, this app seemed to be less popular in other countries (under 10%). Likewise, Italian pupils did not often mention Snapchat as an app of interest (2.4%), but this was much more popular in other countries (29.4–68.1%).

Table 2 also gives details of how pupils reported their parental role in their Internet usage. Regarding whether they told parents about the sites they visit online, on average 19.4% indicated they always tell parents, 22.6% often, 26.4% sometimes, and 31.6% that they do not tell. Regarding parental interest in pupils’ experiences, 11.7% reported that their parents were very interested, 22.8% quite interested, 39.1% a bit interested, but not very much, and 26.4% that they were not interested. There was a moderate correlation between telling parents which sites they visited online, and parental interest in online activities (*r* = .41). Regarding whether parents were concerned about their Internet use, on average 64.3% indicated parents were okay about their Internet use, 6.4% indicated parents were sometimes concerned about the sites pupils visited, 26.0% indicated parents were sometimes concerned about the time they spent on the Internet, and 4.4% indicated that they argued with parents about their Internet use.

### 3.3. Correlates of Internet Usage and Parental Involvement

We explored which demographic characteristics correlated most strongly with frequency of Internet use, time spent on the Internet, parental communication about the sites visited, parental interest in Internet use, and parents’ concern about Internet use. Due to the low number of pupils who indicated gender other than male or female, we had to exclude these from the analyses. In addition, due to the low number of pupils with a physical disability and the relatively low number of pupils with a sensory disability, these two categories were merged and labeled as ‘physical/sensory disability’. Finally, concerning grades, ‘below average’ and ‘average’ were merged, due to the low number of students reporting ‘below average’ scores.

#### 3.3.1. Frequency of Daily Internet Use

There were no unanimous significant correlates across countries (see Table 4). Girls had higher daily Internet usage than boys in Italy and the Netherlands. Majority group members reported higher frequency of daily Internet use in Northern Ireland, Germany, and Italy. Having a learning disability was associated with higher daily Internet usage in England. Pupils with below average or average grades reported higher daily Internet use than pupils with above average grades in Germany and Italy.

#### 3.3.2. Time Spent on the Internet

Again, there was no consistent pattern of correlates across all countries: see Table 5. Girls reported more time spent on the Internet in England and Italy, but boys reported more time on the Internet in Germany. Having a learning disability was associated with more time on the Internet in England. Having fewer books was associated with more time on the Internet in Germany. Pupils with below average or average grades reported spending more time on the Internet than pupils with above average grades, in Germany and the Netherlands.

#### 3.3.3. Communication about Internet Usage

The direction of findings was similar and significant across all five countries for gender (see Table 6). Overall, girls (*M* = 2.59, *SD* = 1.10, *n* = 1079), compared to boys (*M* = 2.08, *SD* = 1.07, *n* = 1372), more often communicated with their parents about visited websites (*t*(2449) = 11.55, *p* < .001, *d* = 1.08). Otherwise, there was no consistent pattern of correlates across all countries. Telling parents was lower for minority group children in Northern Ireland. Pupils with learning disabilities were more likely to tell their parents in Italy.

#### 3.3.4. Parental Interest in Internet Use

The direction of findings was similar and significant across all five countries for gender (see Table 7). Overall, girls (*M* = 2.37, *SD* = 0.93, *n* = 1080), compared to boys (*M* = 2.06, *SD* = 0.96, *n* = 1365), more often said their parents were interested in their Internet use (*t*(2443) = −8.16, *p* < .001, *d* = 0.95). Parental interest was higher for minority group children, and this was significant in Germany and Italy. Pupils with physical/sensory disabilities reported higher parental interest in the Netherlands. Pupils with below average or average grades reported higher parental interest than pupils with above average grades in Germany. Minority group members in Germany and Italy reported higher parental interest in their Internet use in comparison to majority group members.

#### 3.3.5. Parental Concern about Internet Use

The direction of findings was similar across all five countries (see Table 8). Concern was higher for girls, and this was significant in England. Concern was higher for minority group children, significant in Northern Ireland and England. The larger the number of books at home, the more likely parents were to be concerned with their child’s Internet use, and this was significant for Northern Ireland, Italy, and the Netherlands. Pupils with below-average or average grades reported more parental concern than pupils with above-average grades, significant in Italy.

A summary of the significant demographic predictors on Internet use and parental involvement is provided in the Appendix A, Table A1.

## 4. Discussion

The current large-scale survey study was part of the Erasmus+ funded *Blurred Lives* project, which was designed to investigate Internet use amongst adolescents from areas of socioeconomic disadvantage across five European countries: Northern Ireland, England, Germany, Italy, and the Netherlands. The aim of this study was two-fold: (1) to explore adolescents’ self-reported Internet use and their perceived parental involvement in their use; and (2) to explore which demographics were associated most strongly with Internet use and perceived parental involvement. The results provide relevant insight into current Internet use of adolescents of socio-economic disadvantage and suggest points of attention to address in future programs to promote safe and responsible Internet use. An important strength of this study is that we were able to collect a large sample of adolescents from socio-economically disadvantaged backgrounds from five European countries. Consistency in the results between countries argues for the generalizability of such findings.

First, the participating adolescents in these five European countries had very high levels of access to the Internet, both within and outside their home, and over 95% owned a smart phone, despite many of them being from lower socioeconomic backgrounds and attending schools in relatively disadvantaged areas. Over half of these young people accessed the Internet more than 10 times per day, and around two-thirds spent 3 or more hours per day on the Internet. Concerning the popularity of apps, Instagram, YouTube, and Snapchat were used most often. These results are consistent with results from national research reports. Ofcom—which included data from England and Northern Ireland—reported that virtually all adolescents (99% of 12- to 15-year-olds) have Internet access, spending approximately 3 h per day online, with YouTube and Instagram being frequently used apps [3]. In Germany, a total of 96% of the young people surveyed use a smartphone every day or several times a week; the Internet is used by 94% of 12- to 19-year-olds [38]. In Italy, 92.8% of Italian adolescents aged 15–17 years use the Internet on a daily basis, and among users over the age of 14, 91.8% uses smartphones to access the Internet [39]. In the Netherlands, 96% of adolescents are online daily and 99% uses a smartphone for being online [40]. While Internet consumption among adolescents from lower socioeconomic backgrounds is thus very high, it is important to notice is that the regression models indicate that overall, the demographic variables are poor predictors of daily Internet use and frequency of Internet use, since they only accounted for up to 4% of the total variance.

Second, communication with and involvement of parents varied. Less than half of the adolescents in this study often or always told their parents about the websites they had visited; but telling is higher among girls than boys. Less than half reported that their parents were quite or very interested in their activities on the Internet; but again, interest is reported as higher by girls than by boys. Over half of the adolescents reported that their parents were okay about their Internet use. Concern tended to be higher for girls, minority children, those with a disability, those with more books in the home, and those with poorer grades, although there was no consistency across countries.

The perceived lack of involvement by parents has also been reported in other studies and should be considered a source of concern [10,13]. Parental monitoring is known to be an important protective variable as regards cyberbullying and other negative Internet experiences [13]. The literature suggests that parents should find a good balance in allowing autonomy in their child’s Internet use, while on the other hand actively monitoring Internet behavior, setting clear restrictions, and being available when the child encounters negative online experiences [8,10,13,28]. A low level of parental interest is associated with a higher chance for the child to being victimized or become a cyberbully [13]. The current study suggests that boys might be more at risk, since parental involvement was found to be considerably less. The regression models explaining telling parents about Internet use and perceived parental involvement accounted for 2–11% of the total variance, primarily explained by gender differences. This suggests that gender difference with respect to parental interest is of relative importance. There is particular concern in relation to parental interest in Northern Ireland, which had the highest daily time spent online (80.5% of the sample of teenagers spent more than 3 h a day on the Internet) but the lowest reported levels of parental concern about their children’s Internet usage (75.4% reported that their parents were ‘okay’ and so not concerned about their Internet usage).

It should however be noted that children’s reported lack of parental involvement does not per definition mean that parents are not monitoring their child’s online behavior. For example, a recent report investigating online safety from the perspectives of parents and adolescents [41] identified a ‘security perception gap’, in which three-quarters of parents indicate they are confident in keeping their children safe online, but only half of adolescents indicated that they feel supported online by their parents. Such perception of parental involvement is however of major importance in safe Internet use [14]. This suggests that parents should be—together with their child—constructively involved in their children’s Internet use [10,13]. The results of the current study support this view by showing a moderate correlation between adolescents talking about their Internet use and their perceived interest by parents.

### Limitations

There are some limitations to this study which should be considered when interpreting its results. First, the data were all self-reported by the adolescents themselves, and we do not have information from the parent perspective on their levels of concern and involvement. The results of this study, however, are in line with other studies on parental involvement [8,10,13]. Similarly, the popularity of apps was based on an open question; thus, the most popular apps reported are based on recall and do not necessarily reflect objective data. Again, the results do not show any major differences with other studies [3]. Also, as reported above, levels of socioeconomic disadvantage across the five European countries were difficult to compare, leading to inevitable differences among and across the samples. Another limitation is the possibility of selection bias, as schools participated voluntarily. A final limitation is that ethnicity was measured differently across countries because of different ethical regulations regarding registering ethnicity/race. This could affect the results or the interpretation of the results.

## 5. Conclusions

The findings confirm that in all five countries, adolescents have ready access to the Internet, and spend a lot of time on it. This is true for the largely lower-SES pupils in our samples. Parental involvement is varied and is a point of attention for this group. Adolescents indicate that the majority of parents seem to pay little attention to activities on the Internet or are okay with their child’s Internet use. It is true that most parents do not closely monitor the Internet activities of their children, and much Internet use (like social media and gaming) takes place outside the parents’ field of view. However, taking an interest, discussing with adolescents what are the dos and don’ts of Internet use, and making it clear that it is important to discuss and seek help if something bad is happening, are important aspects of parental involvement. A relative lack of this could be a source of concern, especially for boys where parental involvement is considerably less than for girls. It is therefore of great importance that parents receive appropriate training for recognizing problematic Internet use and how to be actively involved in their child’s Internet use in a constructive co-operative way [23,42]. Resources from the UK Safer Internet Centre [43] or Childnet [44] can provide valuable advice and resources for parents in this respect. Furthermore, preventive interventions in schools can address parents more strongly or additional measures for parents can be implemented in schools, for example, to inform parents about their important role in the (appropriate) media use of their children. An evidence-based prevention program such as Medienhelden [45] can be complemented by a school-based measure such as Parent Media [46] to improve parental mediation. An overview of other relevant approaches is reported by Barlett et al. [47].

## Figures and Tables

**Table 1 children-10-01780-t001:** Demographic characteristics.

	Northern Ireland (*n* = 524)	England (*n* = 560)	Germany (*n* = 428)	Italy (*n* = 665)	The Netherlands (*n* = 417)
No. schools	7	5	6	3	9
Age (years) (mean ± SD)	14.92 ± 0.58	14.29 ± 0.77	14.93 ± 0.96	14.93 ± 1.03	14.80 ± 0.73
Gender (%)					
Male	53.8	50.1	54.0	64.4	42.4
Female	42.6	45.3	41.6	31.1	56.4
Other/Unknown/I do not wish to say	3.7	4.6	4.5	4.5	1.2
Urban/rural (*ref. urban*) (%)	86.6	95.5	96.5	76.5	34.5
Ethnicity (% majority)	80.7	16.4	78.3	60.2	89.4
Disability (%)					
None	62.5	82.8	65.3	73.1	72.1
Sensory	15.9	8.6	17.6	11.5	4.1
Physical	6.4	3.6	5.9	3.2	1.7
Learning	26.2	9.0	20.2	18.2	25.3
No. of books (%)					
Just a few	52.3	28.0	26.9	19.3	32.5
About 10 or 20	19.3	22.2	19.4	21.1	20.7
More than 20	28.3	49.8	53.6	59.6	46.9
Grades (%)					
Good/above average	38.5	47.6	26.9	36.2	27.1
About average	50.8	44.8	65.0	57.9	71.0
Not so good/below average	10.7	7.6	8.2	5.9	1.9

**Table 2 children-10-01780-t002:** Internet use.

	Northern Ireland (*n* = 524)	England (*n* = 560)	Germany (*n* = 428)	Italy (*n* = 665)	The Netherlands (*n* = 417)
Device usage (%) ^1^					
Smart phone	96.0	96.2	95.7	97.9	99.3
Tablet	63.9	63.9	48.9	48.5	55.7
Games console	77.6	68.3	67.7	72.3	60.3
Other devices within home	93.8	91.6	94.5	87.3	98.1
Internet access outside home	80.3	78.6	74.0	77.5	85.0
Daily frequency Internet (%)					
I don’t/Maybe once	1.9	4.4	5.7	4.1	0.5
2 to 5 times	12.0	21.3	19.2	18.9	19.1
About 10 times	15.3	20.4	19.0	17.5	26.7
More than 10 times	70.8	54.0	56.1	59.5	53.8
Daily time spent on Internet (%)					
I don’t/less than half an hour	1.7	3.6	1.9	4.1	1.5
About an hour	5.2	9.7	6.6	13.2	7.0
About 2 h	12.5	18.8	15.4	24.2	21.8
3 to 5 h	29.5	30.8	36.2	30.2	43.7
More than 5 h	51.0	37.2	40.0	28.3	26.0
Tell parents about sites visited (%)					
Always	23.8	19.9	11.8	18.5	22.6
Often	27.1	26.3	18.4	21.1	18.5
Sometimes	21.1	27.6	31.4	24.3	29.2
No	27.9	26.1	38.3	36.0	29.7
Parents interest in activities (%)					
Very interested	12.8	16.3	11.8	11.7	4.1
Quite interested	28.4	21.7	18.7	21.6	23.1
A bit interested	35.4	40.0	38.9	35.7	48.2
Not interested	23.3	22.0	30.6	31.0	24.6
Parental concern (%) ^1^					
They are okay about it	75.4	56.7	61.3	59.5	70.9
Concerned about visited sites	7.9	7.3	8.4	5.4	2.9
Concerned about time spent	15.9	32.5	26.3	28.8	25.4
Very worried and we argue about it	2.5	3.5	7.4	6.3	2.0

^1^ Multiple choice questions; the total percentage might exceed 100%. Note. Due to an administrative error in the Italian and English survey, parental concern was asked as a single choice question. As a result, percentages presented might underestimate the actual percentage. It is expected that this is only a minor difference.

**Table 3 children-10-01780-t003:** Apps usage.

	Northern Ireland (*n* = 524)	England (*n* = 560)	Germany (*n* = 428)	Italy (*n* = 665)	The Netherlands (*n* = 417)
Instagram	49.4	50.9	58.4	57.9	77.0
Facebook	58.0	6.3	7.7	8.1	9.8
Twitter	12.0	5.2	5.8	0.9	2.6
Snapchat	60.5	59.1	29.4	2.4	68.1
TikTok	0	0	6.8	1.4	1.4
General mention of social media apps or websites	1.9	1.8	1.2	2.3	1.0
Messaging apps (Snapchat, WhatsApp, iMessage, Facebook Messenger)	25.2	19.3	57.5	36.1	53.7
Video and audio chat apps (e.g., Skype, Facetime, Imo, Houseparty)	1.9	5.4	0.9	0	0.5
TV Streaming Services (e.g., Netflix, SkyGo/BBC iPlayer/ITV Player)	13.2	16.1	12.1	7.7	23.0
Search engines (e.g., Google, Bing, DuckDuckGo)	10.5	17.3	17.1	11.4	9.6
Gaming apps (e.g., Twitch, Discord, Xbox, Steam)	16.0	19.8	23.8	10.5	17.7
Interest apps (e.g., Tumblr, Amino, Episode, Pinterest, Wattpad, Ultimate Guitar)	7.3	10.9	8.9	4.7	10.8
Music apps (e.g., Sound Cloud, Spotify, Deezer)	5.5	5.5	8.2	4.1	10.1
YouTube	47.1	51.6	59.6	46.9	40.8
Homework apps	0.6	12.3	1.2	1.7	9.4
Shopping apps	3.2	5.9	3.7	3.0	2.9
Porn sites and streaming services	1.5	1.3	4.0	7.2	1.2
Other apps	1.7	5.2	3.7	1.7	2.4

**Table 4 children-10-01780-t004:** Results of multiple regression analyses of demographic characteristics on frequency of daily Internet use.

	Northern Ireland (*n* = 467)		England (*n* = 497)		Germany (*n* = 393)		Italy (*n* = 614)		The Netherlands (*n* = 398)	
	*B* [95% CI]	*p*	*B* [95% CI]	*p*	*B* [95% CI]	*p*	*B* [95% CI]	*p*	*B* [95% CI]	*p*
Gender *(ref. male)*	−0.06 [−0.19; 0.07]	.391	0.08 [−0.07; 0.22]	.312	0.06 [−0.10; 0.24]	.447	0.15 [0.01; 0.29]	.042	0.17 [0.01; 0.32]	.037
Ethnicity *(ref. majority)*	−0.18 [−0.34; −0.13]	.034	0.00 [−0.20; 0.20]	.990	−0.22 [−0.42; −0.01]	.040	−0.14 [−0.28; 0.00]	.050	−0.05 [−0.30; 0.20]	.702
Disability *(ref. none)*										
Learning	−0.02 [−0.17; 0.13]	.786	0.51 [0.24; 0.77]	<.001	0.03 [−0.18; 0.24]	.769	0.10 [−0.08; 0.27]	.278	−0.14 [−0.32 0.05]	.140
Physical/sensory	0.04 [−0.13; 0.21]	.634	−0.16 [−0.40; 0.09]	.202	−0.07 [−0.28; 0.15]	.531	0.11 [−0.09; 0.30]	.289	0.12 [−0.21; 0.46]	.464
No. of books	−0.09 [−0.16; −0.01]	.026	0.05 [−0.04; 0.13]	.321	−0.03 [−0.13; 0.07]	.569	0.06 [−0.03; 0.14]	.207	0.04 [−0.05; 0.13]	.420
Grades *(ref. above average)*	−0.10 [−0.24; 0.03]	.141	−0.10 [−0.25; 0.05]	.189	0.26 [0.07; 0.45]	.009	0.14 [0.00; 0.27]	.050	−0.06 [−0.23; 0.12]	.518
*R^2^_adjusted_*	.02	.046	.02	.006	.02	.040	.02	.009	.01	.186

**Table 5 children-10-01780-t005:** Results of multiple regression analyses of demographic characteristics on daily time spent on Internet.

	Northern Ireland (*n* = 469)		England (*n* = 495)		Germany (*n* = 394)		Italy (*n* = 618)		The Netherlands (*n* = 401)	
	*B* [95% CI]	*p*	*B* [95% CI]	*p*	*B* [95% CI]	*p*	*B* [95% CI]	*p*	*B* [95% CI]	*p*
Gender *(ref. male)*	0.08 [−0.90; 0.24]	.373	0.26 [0.08; 0.44]	.006	−0.24 [−0.42; −0.04]	.013	0.39 [0.21; 0.56]	<.001	−0.00 [−0.18; 0.18]	.989
Ethnicity *(ref. majority)*	0.03 [−0.18; 0.23]	.806	0.03 [−0.22; 0.28]	.803	−0.03 [−0.25; 0.20]	.825	0.11 [−0.07; 0.28]	.229	0.27 [−0.01; 0.56]	.060
Disability *(ref. none)*										
Learning	0.16 [−0.04; 0.35]	.107	0.45 [0.12; 0.78]	.007	−0.02 [−0.25; 0.21]	.871	0.14 [−0.08; 0.36]	.205	0.02 [−0.18; 0.23]	.937
Physical/sensory	0.02 [−0.20; 0.23]	.882	−0.18 [−0.48; 0.14]	.250	−0.03 [−0.26; 0.21]	.822	0.22 [−0.03; 0.46]	.084	−0.29 [−0.67; 0.09]	.132
No. of books	−0.08 [−0.18; 0.01]	.092	−0.04 [−0.15; 0.07]	.503	−0.13 [−0.24; −0.02]	.026	−0.07 [−0.18; 0.03]	.178	0.05 [−0.06; 0.15]	.368
Grades *(ref. above average)*	0.03 [−0.14; 0.20]	.694	0.09 [−0.10; 0.28]	.346	0.34 [0.14; 0.55]	.001	0.04 [−0.13; 0.21]	.660	0.23 [0.03; 0.42]	.026
*R^2^_adjusted_*	.00	.370	.02	.009	0.04	<.001	.04	<.001	.01	.078

**Table 6 children-10-01780-t006:** Results of multiple regression analyses of demographic characteristics on frequency of telling parents about visited sites.

	Northern Ireland (*n* = 465)		England (*n* = 494)		Germany (*n* = 394)		Italy (*n* = 623)		The Netherlands (*n* = 400)	
	*B* [95% CI]	*p*	*B* [95% CI]	*p*	*B* [95% CI]	*p*	*B* [95% CI]	*p*	*B* [95% CI]	*p*
Gender *(ref. male)*	0.74 [0.54; 0.94]	<.001	0.35 [0.17; 0.54]	<.001	0.40 [0.20; 0.60]	<.001	0.37 [0.18; 0.56]	<.001	0.76 [0.54; 0.97]	<.001
Ethnicity *(ref. majority)*	−0.35 [−0.61; −0.10]	.006	−0.16 [−0.41; 0.10]	.224	0.08 [−0.16; 0.33]	.507	0.05 [−0.14; 0.23]	.626	−0.27 [−0.62; 0.07]	.121
Disability *(ref. none)*										
Learning	0.00 [−0.23; 0.24]	.980	−0.43 [−0.38; 0.30]	.805	0.20 [−0.05; 0.46]	.113	0.28 [0.05; 0.52]	.018	0.06 [−0.19; 0.30]	.664
Physical/sensory	−0.02 [−0.29; 0.24]	.872	−0.14 [−0.45; 0.16]	.358	−0.17 [−0.43; 0.08]	.185	−0.05 [−0.31; 0.22]	.736	−0.14 [−0.59; 0.32]	.560
No. of books	−0.10 [−0.21; 0.02]	.099	0.08 [−0.03; 0.20]	.155	0.09 [−0.03; 0.22]	.133	0.07 [−0.04; 0.19]	.208	0.00 [−0.12; 0.13]	.954
Grades *(ref. above average)*	−0.07 [−0.27; 0.14]	.511	−0.03 [−0.22; 0.16]	.773	−0.12 [−0.35; 0.12]	.328	−0.14 [−0.33; 0.04]	.131	−0.12 [−0.36; 0.12]	.332
*R^2^_adjusted_*	.11	<.001	.03	.003	.03	.005	.03	<.001	.10	<.001

**Table 7 children-10-01780-t007:** Results of multiple regression analyses of demographic characteristics on parental interest on Internet use.

	Northern Ireland (*n* = 466)		England (*n* = 489)		Germany (*n* = 393)		Italy (*n* = 613)		The Netherlands (*n* = 400)	
	*B* [95% CI]	*p*	*B* [95% CI]	*p*	*B* [95% CI]	*p*	*B* [95% CI]	*p*	*B* [95% CI]	*p*
Gender *(ref. male)*	0.36 [0.18; 0.54]	<.001	0.20 [0.02; 0.37]	.027	0.24 [0.04; 0.43]	.018	0.32 [0.16; 0.49]	<.001	0.46 [0.30; 0.61]	<.001
Ethnicity *(ref. majority)*	−0.03 [−0.26; 0.19]	.769	−0.17 [−0.41; 0.07]	.160	0.28 [0.04; 0.52]	.023	0.19 [0.03; 0.35]	.024	0.12 [−0.13; 0.36]	.350
Disability *(ref. none)*										
Learning	0.10 [−0.11; 0.30]	.372	−0.02 [−0.34; 0.31]	.928	0.07 [−0.18; 0.31]	.600	−0.02 [−0.23; 0.18]	.827	−0.04 [−0.22; 0.14]	.636
Physical/sensory	−0.21 [−0.45; 0.02]	.076	−0.27 [−0.56; 0.02]	.070	−0.01 [−0.26; 0.24]	.920	0.07 [−0.16; 0.30]	.574	0.33 [0.00; 0.66]	.047
No. of books	0.05 [−0.05; 0.16]	.327	0.05 [−0.06; 0.15]	.402	0.02 [−0.10; 0.14]	.723	0.10 [−0.01; 0.20]	.062	−0.01 [−0.10; 0.08]	.821
Grades *(ref. above average)*	−0.04 [−0.22; 0.14]	.681	−0.17 [−0.35; 0.01]	.065	−0.25 [−0.48; −0.03]	.026	−0.14 [−0.30; 0.02]	.086	0.08 [−0.09; 0.25]	.348
*R^2^_adjusted_*	.03	.001	.02	.013	.03	.012	.03	<.001	.08	<.001

**Table 8 children-10-01780-t008:** Results of binary regression analyses of demographic characteristics on parental concern on Internet use.

	Northern Ireland (*n* = 468)		England (*n* = 493)		Germany (*n* = 392)		Italy (*n* = 614)		The Netherlands (*n* = 399)	
	*OR* [95% CI]	*p*	*OR* [95% CI]	*p*	*OR* [95% CI]	*p*	*OR* [95% CI]	*p*	*OR* [95% CI]	*p*
Gender *(ref. male)*	1.19 [0.77; 1.85]	.438	0.92 [0.64; 1.33]	.667	0.74 [0.49; 1.14]	.175	0.56 [0.39; 0.79]	.001	1.15 [0.74; 1.80]	.539
Ethnicity *(ref. majority)*	0.54 [0.32; 0.90]	.018	0.53 [0.32; 0.89]	.016	0.66 [0.40; 1.10]	.109	0.77 [0.54; 1.09]	.136	0.60 [0.30; 1.19]	.141
Disability *(ref. none)*										
Learning	1.18 [0.70; 1.99]	.536	1.44 [0.73; 2.83]	.289	1.09 [0.64; 1.84]	.757	1.29 [0.83; 2.01]	.259	1.01 [0.60; 1.69]	.985
Physical/sensory	0.59 [0.34; 1.00]	.052	0.91 [0.50; 1.65]	.749	0.64 [0.38; 1.08]	.092	0.79 [0.49; 1.29]	.350	1.01 [0.40; 2.56]	.986
No. of books	0.60 [0.47; 0.77]	<.001	0.92 [0.74; 1.15]	.452	0.92 [0.71; 1.18]	.507	0.76 [0.61; 0.95]	.016	0.68 [0.53; 0.89]	.005
Grades *(ref. above average)*	0.74 [0.47; 1.17]	.202	0.92 [0.63; 1.33]	.652	1.23 [0.77; 1.98]	.391	0.63 [0.45; 0.90]	.011	1.21 [0.74; 1.97]	.443
*Nagelkerke R^2^*	.09	<.001	.02	.185	.03	.163	0.05	<.001	.04	.098

## Data Availability

The data presented in this study are openly available in Open Science Framework at https://doi.org/10.17605/OSF.IO/PCFUE, reference number [48].

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
