# Peer review of "Internet Use and Perceived Parental Involvement among Adolescents from Lower Socioeconomic Groups in Europe: An Exploration"

_children, 2023, doi:10.3390/children10111780_

Round 1

Reviewer 1 Report

Comments and Suggestions for Authors

Dear Authors,

The paper describes internet use among adolescents, and parental involvement/concern about it, from areas of socio-economic disadvantage across 5 European countries. This study focuses on a critical phenomenon, thus internet use among adolescence. The manuscript is well-written. However, some weaknesses need to be addressed and improved before publishing. Next, I will share some concerns and questions about specific parts of the article.

Introduction

1.     Although these are important variables for the aim of the study and the interpretation of the results, I found that the literature on the relationship between internet use and demographic characteristics (especially gender and ethnicity) could be improved by explaining the relationships in more detail. In addition, in line with the aims, it might be useful to expand the literature on parental involvement/concern. Finally, the study is highly valuable for inclusion of different countries, however, this aspect is addressed marginally in the introduction. I would expand the literature on differences between countries regarding internet use. In fact, the results would be more related to the existing literature if the latter were enriched.

2.     Line 87-89 - “However we further consider aspects of parental involvement in their child’s internet use”: aspects of parental involvement described in this way suggests the presence of questions addressed personally to parents; perhaps it is more appropriate, in this case, to refer to 'children's perceptions' of parental involvement in their use of the Internet.

Method

Participants and procedure: In particular, lines 103-104 specified that participants aged 13 and 17-18 were also included, while lines 105-107 reported that those who did not meet the age criterion were excluded from the analyses. I would like to clarify this point better: were participants aged 13 and 17-18 included in the analyses? If so, how many participants of this age are involved? The article mentions a sample of 14-16 years, if younger or older participants were included, this wording should be revised.

Discussion

1.     Discussion: The discussions should not be a summary of the results. it is absolutely necessary to discuss the data on the basis of the existing literature.

2.     Limitations: The interpretation of the results could be influenced by the necessary difference between countries in considering variables (e.g. the manner in which ethnicity is coded). This can be discussed as a limitation.

3.     Conclusion: Please argue the practical and applicative implications of this research, for example in relation to possible interventions that could be implemented in and out of school.

Author Response

Comments and Suggestions for Authors

The paper describes internet use among adolescents, and parental involvement/concern about it, from areas of socio-economic disadvantage across 5 European countries. This study focuses on a critical phenomenon, thus internet use among adolescence. The manuscript is well-written. However, some weaknesses need to be addressed and improved before publishing. Next, I will share some concerns and questions about specific parts of the article.

Introduction

  1. Although these are important variables for the aim of the study and the interpretation of the results, I found that the literature on the relationship between internet use and demographic characteristics (especially gender and ethnicity) could be improved by explaining the relationships in more detail. In addition, in line with the aims, it might be useful to expand the literature on parental involvement/concern. Finally, the study is highly valuable for inclusion of different countries, however, this aspect is addressed marginally in the introduction. I would expand the literature on differences between countries regarding internet use. In fact, the results would be more related to the existing literature if the latter were enriched.

Thank you for your constructive review and valuable suggestions for improvement. We have expanded the introduction section and added more detail to the role of parental involvement (lines 79-95). The literature on the factor associated with internet use are presented in a separate section and discusses additional literature (lines 97-129).

Concerning literature discussion differences between countries, we decided to expand on this in the discussion section. We felt that this was a better place to compare the results of our study to the results of other research.

  1. Line 87-89 - “However we further consider aspects of parental involvement in their child’s internet use”: aspects of parental involvement described in this way suggests the presence of questions addressed personally to parents; perhaps it is more appropriate, in this case, to refer to 'children's perceptions' of parental involvement in their use of the Internet.

Thank you for this suggestion. Indeed, the wording can me misinterpreted. We have critically looked at the description of the aim of the paper and decided to write it out more clearly. We also decided to be more consistent in referring to the target group, and now refer to them as adolescents (lines 146-160):

‘While we have some insight in adolescent’s internet use and relevant factors related to adolescent’s internet use, there is little insight in how this applies for the at-risk group of adolescents of socioeconomic disadvantage. Furthermore, we have little insight in how this group perceives parental involvement in their internet use and which demographics are most importantly related to this perception. Therefore, the aim of this explorative study is twofold. First, we will describe the extent and frequency of internet use in areas of socioeconomic disadvantage – including popular apps and sites, as well as perceived parental involvement in adolescents’ internet use. Relevant demographics, including gender, ethnicity, disability, school performance and economic status will also be included in this description. Second, we aim to explore which demographics correlate most strongly with internet use and perceived parental involvement. In this large-scale study reporting data from five European countries, we specifically looked for consistent and inconsistent findings across countries. However, we did not use statistical tests to examine country differences, since specific measurement-theoretical conditions should be in place for such comparisons to be made [33,34].’

Method

Participants and procedure: In particular, lines 103-104 specified that participants aged 13 and 17-18 were also included, while lines 105-107 reported that those who did not meet the age criterion were excluded from the analyses. I would like to clarify this point better: were participants aged 13 and 17-18 included in the analyses? If so, how many participants of this age are involved? The article mentions a sample of 14-16 years, if younger or older participants were included, this wording should be revised.

Thank you for pointing this out. Indeed, the inclusion criteria could be specified better. We have rewritten this section (lines 176-188):

‘Informed consent from a school representative was required in order to collect data within schools. Furthermore, parents were informed about the research and – depending on the ethical guidelines of the country – had to provide active informed consent. Parents could withdraw permission for their child to participate at any time. Students had to sign informed consent before filling in the questionnaire. In total 2,655 students filled in the questionnaire (age range = 10-26 years). We primarily aimed for grade years containing 14- to 16-year-olds. In total, 2,382 students fell within this age range (89,7%). Since we recruited at grade level, some early students (age = 13, n = 137) and late students or repeaters (age = 17-18, n = 75) also filled in the questionnaire. We decided to include these students in the analyses. Sixty-one pupils were removed from the dataset for not meeting this age criteria of 13-19 years (age < 13, n = 7; or age > 18, n = 9) or information on age was not entered correctly (i.e., age = 1-4, n = 17) or missing (n = 28), resulting in a sample size of 2,594 students. There were no other exclusion criteria besides the age criterium.’

Discussion

  1. Discussion: The discussions should not be a summary of the results. it is absolutely necessary to discuss the data on the basis of the existing literature.

Indeed, the discussion section primarily summarized the results. We have rewritten the complete discussion section (see lines 407-482), reflecting on the internet usage, and most importantly discussing the role of parental involvement in the internet use of adolescents of socioeconomic disadvantage.

  1. Limitations: The interpretation of the results could be influenced by the necessary difference between countries in considering variables (e.g. the manner in which ethnicity is coded). This can be discussed as a limitation.

Thank you for your suggestion. Indeed, the different ways in which ethnicity was measured should be reported as a limitation. We therefore have included the following lines in the limitation section (lines 480-482):

‘A final limitation is that ethnicity was measured differently across countries because of different ethical regulations regarding registering ethnicity/race. This could affect the results or the interpretation of the results.’

  1. Conclusion: Please argue the practical and applicative implications of this research, for example in relation to possible interventions that could be implemented in and out of school.

Based on the results of the study, we now provided important recommendations for future interventions in the conclusion section (lines 484-504).

Reviewer 2 Report

Comments and Suggestions for Authors

The article addresses an important topic, the internet use of young people. The literature review is sound, highlights all relevant aspects of the researched topic through references mostly form the past 5 years . The data introduced are well structured and well presnted, but it is not clear, why the very target group of disadvantaged teenagers have been selected and what the authors have expected to gain by selecting this very target group. Do they suppose/propose being not so dfferent, or on the contrary very different from their peers with more stable backgrounds? It would be lovely to have a clear explanation regarding this, and of cours reflections related to it later in.

Comments on the Quality of English Language

There are some typos and grammatical mistakes that need correcting.

Author Response

Comments and Suggestions for Authors

The article addresses an important topic, the internet use of young people. The literature review is sound, highlights all relevant aspects of the researched topic through references mostly form the past 5 years. The data introduced are well structured and well presented, but it is not clear, why the very target group of disadvantaged teenagers have been selected and what the authors have expected to gain by selecting this very target group. Do they suppose/propose being not so different, or on the contrary very different from their peers with more stable backgrounds? It would be lovely to have a clear explanation regarding this, and of course reflections related to it later in.

Thank you for your review. In the introduction, we now explicitly stated that problematic internet use is a risk factor in particular for adolescents from areas of socio-economic disadvantage (lines 63-67) and referred to several relevant research articles supporting this claim references:

Socially disadvantaged youth is a vulnerable group with respect to risky or addictive behavior, and it is therefore important take into account specific factors linked to youth in structurally disadvantaged regions when developing health programs [8]. With respect to internet use, several studies point out that problematic internet use is a risk factor in particular for adolescents from areas of socio-economic disadvantage [8–10].

We reflected on the role of parental support for this group in the discussion section.

Comments on the Quality of English Language

There are some typos and grammatical mistakes that need correcting.

We have critically reread the manuscript with special attention to English language. We made various changes in readability and language use throughout the whole manuscript.

Reviewer 3 Report

Comments and Suggestions for Authors

Thank you very much to the editors for the opportunity to review the article – Internet use among young people from lower socioeconomic groups in Europe.

I am commenting on the text below

Introduction

- References should be written in square brackets, please correct

Materials and methods

- What were the exclusion criteria for the study?

- Whether the research has been approved by the ethics committee - I do not see the approval number of the committee

- I don't understand the point of using yellow for some results. Please explain this matter

Discussion

The discussion needs extensive improvement. In my opinion, it is more of a conclusion

Author Response

Comments and Suggestions for Authors

Thank you very much to the editors for the opportunity to review the article – Internet use among young people from lower socioeconomic groups in Europe.

I am commenting on the text below

Introduction

  • References should be written in square brackets, please correct

Thank you for pointing this out. We have now have closely followed the journal guidelines, including the reference style.

Materials and methods

What were the exclusion criteria for the study?

Please also see our response to reviewer one on the inclusion and exclusion criteria. Besides the age criterium, no other exclusion criterium were set. We now explicitly stated this in the manuscript (line 176-188).

Whether the research has been approved by the ethics committee - I do not see the approval number of the committee

We made a general statement about this in the manuscript (lines 164-165):

The study was conducted in agreement with ethical guidelines for protecting human participants. All partners obtained approval of their local Ethics Committee.

At the end of the manuscript, we provided detailed information on the ethics approval for each respective country (lines 522-532):

‘Institutional Review Board Statement: The study was conducted in accordance with the Declaration of Helsinki, and approved by the university’s ethics committee of each respective country: approval of the ethics committee at the Open Universiteit in the Netherlands was received on June 6, 2018, under reference number: U2018/03921/MQF; approval of the Ethic committee of the University of Bologna in Italy was received on May 5, 2018, under reference number: n.68251; approval of the Ethics Committee of the Department of Education & Psychology at Freie Universität Berlin in Germany was received on October 16, 2018, under reference number: 191/2018; approval of the Research Ethics & Integrity Sub-Committee in the United Kingdom was received on June 29, 2018, under reference number: 1393; approval of the Research and Ethics Committee, Stranmillis University College, Belfast, Northern Ireland was received on March 12, 2018, under reference number: 2018-03PURDY.’

I don't understand the point of using yellow for some results. Please explain this matter

Our apologies for this unclarity. The highlighted results are the significant results. We highlighted them as they helped with the interpretation of the data but forgot the unmark them in the submitted manuscript. We removed this marking.

Discussion

The discussion needs extensive improvement. In my opinion, it is more of a conclusion

Thank you for this suggestion. Please also see our response to reviewer 1. We agree that the discussion should in more detail discuss the results. We therefore have rewritten the complete discussion section.